# Evaluation of Salivary Biochemistry in Dogs with and without Plaque, Calculus, and Gingivitis: Preliminary Results

**DOI:** 10.3390/ani12091091

**Published:** 2022-04-22

**Authors:** Anna Perazzi, Rebecca Ricci, Barbara Contiero, Ilaria Iacopetti

**Affiliations:** Department of Animal Medicine, Production and Health, University of Padua, 35020 Padua, Italy; anna.perazzi@unipd.it (A.P.); barbara.contiero@unipd.it (B.C.); ilaria.iacopetti@unipd.it (I.I.)

**Keywords:** dog, saliva, biochemistry, plaque, calculus, gingivitis

## Abstract

**Simple Summary:**

Periodontal disease is one of the most prevalent disorders observed in dogs requiring primary-care veterinary services. Traditional methods for its diagnosis involve clinical measurements requiring sedation or general anesthesia. The aim of this study was to evaluate whether quantified salivary biochemistry parameters can be used as markers of periodontal disease in dogs. Seventy-nine dogs were allocated into three groups according to the severity of periodontal disease: none (Group 1), moderate (Group 2), and severe (Group 3). A blood sample and a saliva sample were collected from each dog to quantify biochemical parameters that included alpha-amylase, lysozyme, lactate dehydrogenase (LDH), calcium, and phosphorus. LDH and phosphorus showed the highest values in Group 3 whereas calcium, amylase, and lysozyme did not differ among groups. The salivary phosphorus cut-off value of 4.04 mg/dl that was established signified that above such value, periodontal disease could be predicted with fairly high probability. Although further studies are needed to confirm these preliminary results, this study offers a valid starting point for further investigation into the role of saliva as a non-invasive tool for detecting periodontal disease in dogs.

**Abstract:**

This study evaluated whether salivary alpha-amylase, lysozyme, lactate dehydrogenase (LDH), calcium, and phosphorus can be used as markers of periodontal disease in dogs. Plaque, calculus, and gingivitis indexes were used to allocate 79 dogs in three groups: none (Group 1), moderate (Group 2), and severe (Group 3) periodontal disease. A blood sample and a saliva sample were collected from each dog to quantify biochemical parameters. LDH and phosphorus showed the highest values in Group 3 (LDH: Group 1, 2559.85 ± 676.95; vs. Group 2: 1636.76 ± 597.36 vs. Group 3: 4099.18 ± 545.45 U.I./l, *p* = 0.016; phosphorus: Group 1, 3.02 ± 0.76 vs. Group 2: 5.34 ± 0.67 vs. Group 3: 5.85± 0.61 mg/dl, *p* = 0.049) whereas calcium, amylase, and lysozyme did not differ among groups. A ROC curve analysis permitted the establishment of a salivary phosphorus cut-off value of 4.04 mg/dl, above which periodontal disease could be predicted (70% sensitivity (95% CI: 50.6–85.3%) and 72.41% specificity (95% CI: 52.8–87.3%)). Only salivary calcium was higher in males; no other salivary parameters appeared affected by gender or age. Although further results on a larger population are needed, this study shows that salivary LDH and phosphorus increase in dogs affected by plaque, supragingival calculus, and gingivitis, and this suggests their potential use as markers of periodontal disease in canine species.

## 1. Introduction

Plaque, calculus, and gingivitis are common features of periodontal disease, a chronic inflammatory disorder affecting wide ranges of the population in both humans and dogs [1,2,3,4,5]. More specifically, the incidence and severity of periodontal disease in dogs has been shown to increase with age. Furthermore, small breeds of dog, brachycephalic breeds, and dogs with tooth overcrowding appear to be especially vulnerable, and in some cases, extracting the affected teeth is the only effective therapy possible [4,6,7].

Periodontal disease derives from the accumulation of plaque, a complex mixture of organisms in a matrix of salivary glycoproteins and extracellular polysaccharides [2,8] on tooth surfaces and from the inflammatory and immune responses in tooth support tissues [1,3]. The oral bacterial microflora in dogs and cats is composed of aerobic, facultative, or strictly anaerobic bacteria [9]. To date, around 500 bacterial species have been recognized in the mouth of dogs and cats, both healthy and with periodontal disease [8]. A process of mineralization (calcification) that leads to calculus formation begins in plaque shortly after its formation. Anaerobic bacteria adhering to plaque intensify fermentation processes, increase acidity in the mouth, and accelerate the enamel demineralization process [8,9,10].

Traditional methods for diagnosing periodontal disease involve clinical measurements that assess probing depth, clinical attachment levels, plaque index, gingival index, gingival bleeding on probing, furcation involvement, tooth mobility, and radiographic assessments that are poorly tolerated by canine patients, and for this reason require sedation or general anesthesia with which owners are reluctant to comply [1,8,11].

Identifying non-invasive methods that both assess the presence and severity of periodontal disease and identify patients at risk deserves attention. To this end, research on humans that has focused on evaluating salivary compounds and the many biomarkers contained in saliva holds promise in diagnosing periodontal disease. Saliva is also easily accessible and can be sampled repeatedly using non-invasive methods [1]. Together with mucin and total proteins, salivary α-amylase is known as an important biochemical parameter of inflammation of the periodontium in human patients [1,12].

One of the most abundant enzymes in human saliva, alpha-amylase, together with salivary ions and salivary proteins, takes part in forming the acquired enamel pellicle and is also involved in the colonization and metabolism of bacteria responsible for plaque formation [13,14]. Alpha-amylase is one of the inflammatory salivary biomarkers used most frequently in recent decades [15].

Lysozyme is part of the innate salivary defense mechanism. Along with numerous proteins and peptides such as lactoferrin, lactoperoxidase, statherin, histatins, and secretory immunoglobulin A, it performs antimicrobial activity, controls microbial overgrowth, reduces the number of bacteria in dental biofilm, modifies bacterial metabolism, and decreases colonization [13,16].

In human patients, higher levels of salivary lactate dehydrogenase (LDH) have been linked to the inflammation and damage of oral tissues commonly caused by gingivitis and periodontitis [17]. Significant increases in salivary total LDH and LDH isoenzyme activity levels have been observed in patients with oral submucous fibrosis (OSMF) and oral squamous cell carcinoma (OSCC), thus indicating LDH as a potential marker in diagnosing these malignant disorders [17,18]. The relationship between salivary calcium and phosphorus concentration and the degree of gingival and periodontal inflammation has also been studied in human patients [19].

Concentrations of certain salivary parameters related to periodontal disease in humans, namely α-amylase, lysozyme, lactate dehydrogenase, calcium, and phosphorus, have been quantified in a population of healthy dogs [20]. Whether or not salivary biochemistry is affected by periodontal disease in this latter species has not yet been investigated, however. The aim of this study was to quantify the salivary concentration of α-amylase, lysozyme, lactate dehydrogenase, calcium, and phosphorus in an unanesthetized dog population affected by plaque, supragingival calculus, and gingivitis and to evaluate whether these parameters can be used as markers of periodontal disease in dogs.

## 2. Materials and Methods

Dog population—In order to be included in the study, dogs had to be healthy. With this aim, each individual was submitted to a general physical examination and its hematological and serum biochemical analytes had to fall within physiological ranges. They also had to have been fed commercial dry pet food. Dogs suffering from known major systemic disease or oral disease other than periodontal disease such as trauma, hyperplastic or neoplastic lesions were excluded from the study, as were those that had received periodontal therapy or special dental care or received drugs in the past 6 months; additionally, scarcely collaborative or biting dogs were excluded. The study included both entire and sterilized owned dogs of various age, breed, body weight, and either gender. The owner’s written consent was obtained prior to the dogs’ enrollment. The study was approved by the Animal Welfare Committee of the University of Padua (Authorization number n°#71/2015). The entire population was recruited from patients of the Veterinary Teaching Hospital of the University of Padua and consisted of both dogs with no signs of periodontal disease and dogs affected by varying degrees of plaque, supragingival calculus, and gingivitis.

Periodontal disease scoring—The severity of periodontal disease was scored by giving each dog an oral-dental examination without sedation. Plaque, supragingival calculus, and gingivitis were scored from 0 to 4 as described by Warrick et al. [5], modified as reported in Table 1.

Probing and radiographic evaluation were not performed in order to avoid these clinical evaluations in non-sedated animals [5]. The scoring system for plaque, supragingival calculus, and gingivitis was adopted for canines, premolars, and molars, both maxillary and mandibular. Each dog received an average score for each parameter analyzed. This score was the mean score of all teeth assessed. Based on the scores obtained, the population was divided into the 3 groups described in Table 2: Group 1 included dogs with no periodontal disease; Group 2, dogs with moderate periodontal disease; Group 3, dogs with severe periodontal disease. In order to be assigned to one of the three groups, individuals were required to show an average score in the range indicated in Table 2 for at least 2 out of the 3 parameters (plaque, calculus, and gingivitis). Subjects that failed to meet this requirement were excluded.

Saliva and blood collection and analysis—Both saliva and blood samples were collected in the morning (between 9:00 and 12:00 a.m.) and processed and analyzed as previously reported by Iacopetti et al. [20]. Briefly, dogs were prohibited from eating 12 h prior to sampling and water was removed at least 1 h previously. Two dental cotton rolls (Salivette^®^ tubes, SARSTEDT AG & Co., D-51582 Nümbrecht, Germany) were inserted in the dog’s oral cavity one at a time and the dog was allowed to chew for 1 min. Each cotton roll was immediately centrifuged for two minutes at 1000× *g* (Labofuge 400, Heraeus Holding, Hanau, Germany) and the saliva samples obtained from each dog were pooled together. The choice of using two cotton rolls stemmed from the need to obtain a sufficient amount of saliva from each dog. Saliva analysis conducted using a BT1500 automated chemistry analyzer included the quantification of amylase, LDH, lysozyme, Ca, and P. Equipment calibration was the same as adopted for blood collection.

After saliva collection, blood samples were taken from each fasted dog by venipuncture from the cephalic vein and placed in evacuated plastic tubes containing either a coagulation accelerator or EDTA for biochemical and hematological analysis. The EDTA blood samples were analyzed using ADVIA 120 Hematology Systems—Siemens Healthcare (Siemens Italia, 20128 Milano, Italy) equipped with Veterinary Software version 3.18.0-MS. A blood smear was performed for each sample to confirm ADVIA data. The hematological analytes included packed cell volume (PCV), platelet count (PLT), leukocyte count (WBC), and the relative and absolute numbers of neutrophils, lymphocytes, monocytes, eosinophils, basophils, and large unstained cell (LUC) counts and values for mean platelet volume (MPV), large platelet count, and platelet clumps. For serum biochemical analysis, coagulated samples were centrifuged (Labofuge 400, Heraeus Holding, Hanau, Germany) at 1750× *g* for 10 min at room temperature; serum was separated and analyzed immediately using a BT1500 automated chemistry analyzer. Magnesium, alkaline phosphatase, aspartate aminotransferase, alanine aminotransferase, gamma-glutamyltransferase, LDH, creatinine, urea, total protein, globulins, cholesterol, triglycerides, glucose, and creatine kinase were analyzed in serum, together with the analytes quantified in saliva.

Statistical analysis—Descriptive statistics of salivary variables were grouped by dog age, gender, reproductive condition, and size. Differences in demographic variables were assessed using either chi-square Fisher’s exact test or one-way ANOVA for non-continuous and continuous variables, respectively. The post hoc pairwise comparisons among Groups were calculated using Marasquilo approach or Bonferroni correction. The normally distributed salivary variables were analyzed using a linear model that included the fixed effects of class of age (≤2 years, 2–7 years, >7 years), gender (M vs. F), sterilization (entire vs. neutered), severity of periodontal disease (Group 1 vs. Group 2 vs. Groups 3), and interaction between gender and sterilization. The hypothesis of the linear model on the residuals was visually inspected. Non-normally distributed data were log transformed. Data were reported as least square means ± standard error. The log-transformed estimates were back-transformed to obtain results in original scale [21].

The non-parametric test was applied whenever log transformation failed to achieve normality. ROC curve analysis was used to determine the cut-off value between healthy dogs and diseased dogs for the salivary variables in which the effect of the severity of periodontal disease was statistically significant. The values of the area under the curve (AUC) as a criterion of the accuracy of the variables tested were defined as low (0.5–0.7), moderate (0.7–0.9), or high (>0.9). All analyses were performed using statistical software packages (SAS v. 9.3 and MedCalc v. 12.4.0). Furthermore, an ROC curve analysis was built for the salivary variables that appeared to be affected by periodontal disease in order to establish the cut-off that discriminates healthy dogs from those with periodontal disease.

## 3. Results

Inclusion criteria were met and blood parameters were not indicative of disease in seventy–nine dogs (data not shown), which were therefore recruited in the study. Among them, 29 had a healthy oral cavity condition with no signs of plaque, supragingival calculus, or gingivitis (Group 1); 15 showed moderate periodontal disease (Group 2) and had severe periodontal disease (Group 3). Table 3 lists the characteristics of the study population by periodontal status.

The age of dogs ranged from 10 months to 15 years. Most dogs with a healthy oral condition (Group 1) were less than 2 years old, whereas the worst oral condition (Group 3) was observed in individuals more than 7 years old. No differences in oral condition were seen between females and males, whereas a more significant number of individuals with no periodontal disease were not sterilized. The population was mainly composed of small-size (31) and medium-size (35) dogs distributed among the three periodontal status classes. There were more small-size and medium-size dogs than large-size dogs in Groups 2 and 3 (61% and 68% vs. 54%). In general, the dogs were in satisfactory physical condition (BCS: 2.95 ± 0.5) with no differences detected among the three groups (*p* = 0.842).

The salivary biochemistry values obtained in Group 1, Group 2 and Group 3 are shown in Table 4.

P and LDH were influenced by the grade of the periodontal disease, with the highest value observed in Group 3, whereas calcium, amylase and lysozyme did not differ among groups. Because P was the only parameter showing a significant growth trend in proportion to the increasing severity of oral condition, a ROC curve analysis (Figure 1) was performed on the results that established 4.04 mg/dl as the P cut-off value above which a dog is considered affected by periodontal disease with a sensitivity of 75.5% (95% CI: 61.1–86.7%), a specificity of 72.4% (95% CI: 52.8–87.3%), and an under curve area (AUC) of 0.75, which corresponds to a moderately accurate test, according to Swets (1998) [22].

In this study population, no effect of age or gender was observed on salivary parameters (Table 5 and Table 6) except for calcium, which was significantly higher in male dogs.

No correlation was found between the analytes quantified in saliva and the same ones measured in serum (data not shown).

## 4. Discussion

The aim of this study was to assess whether or not salivary biochemistry is affected by different grades of plaque, supragingival calculus, and gingivitis in a client-owned dog population affected by periodontal disease.

Seventy-nine healthy dogs were included in the study. Among them, the majority showed either a healthy oral cavity condition, with no signs of plaque, supragingival calculus or gingivitis (Group 1) or severe periodontal disease (Group 3). It is not surprising that most dogs in Group 1 were less than 2 years old, nor that the worst oral condition was mostly observed in dogs more than 7 years old. Periodontal disease is a frequently observed oral disease in dogs, with incidence and severity increasing with age [6]; Marshall et al. also demonstrated that as dogs age, they progress towards periodontitis more quickly than younger dogs when efforts at maintaining oral hygiene are stopped. Just as no gender predisposition has been previously documented elsewhere [4], it was not observed here.

In this study population, more small-size and medium-size dogs than large-size dogs were present in the groups with periodontal disease. Smaller breeds of dog have been reported to be more susceptible and to present earlier onsets of periodontal disease than larger breeds. Among the former, Yorkshire Terriers, Toy and Miniature Poodles, Dachshunds, Cocker Spaniels, and Jack Russell Terriers are the breeds most affected [6]. The high susceptibility to periodontal disease among these breeds may be explained by both genetic predisposition and the overcrowding of teeth and malocclusions that may cause tooth rotation or overlapping cusps, thus leading to the entrapment of food and other debris. The identification of a fairly high number of medium-size dogs with periodontal disease may be due to the fact that the authors strived to recruit individuals that allowed withdrawals of sufficient amounts of saliva for analyses. The distribution of the population by body size is therefore biased by such aim.

Periodontal disease is the most common oral disease in dogs, with prevalence estimates ranging between 44 and 100%; moreover, it is one of the most prevalent disorders reported in dogs receiving primary-care veterinary services in England [6]. Traditional methods for diagnosing periodontal disease involve clinical probing and radiographic assessments often poorly tolerated by dogs, who must therefore undergo anesthesia procedures. For such reason, finding a quicker, non-invasive ways to identify periodontal disease in dogs must rightly be given attention.

Various studies aimed at quantifying compounds in human oral fluids to assess the presence and severity of periodontal disease and patients at risk have been conducted [1]. The use of saliva as a diagnostic fluid has also been motivated in large scale screening and epidemiologic studies in recent decades by its proven effectiveness in monitoring general health, detecting the onset of disease, and documenting its progression [23]. Whole saliva can be collected non-invasively even by individuals with limited training [20,24].

In this study, calcium, phosphorus, lysozyme, LDH, and amylase were measured in saliva of both healthy dogs (Group 1) and in dogs with different grades of periodontal disease (Groups 2 and 3). Only LDH and phosphorus were seen to be affected by oral condition, and Group 3 showed the highest values.

Salivary lactate dehydrogenase enzyme is the salivary component subjected to the greatest study and has been proved to be a specific indicator of oral health and tissue integrity [17,25]. Studies by De La Peña et al. and Todorovic et al. demonstrate that LDH enzyme activities in saliva were significantly higher in patients with periodontal disease than in healthy ones [26,27]. Moreover, Numabe et al. showed that LDH activity was reduced after patients underwent periodontal therapy [25]. In canine medicine, the only data available on salivary LDH have been retrieved from a population of healthy individuals [20]. In this study, although the LDH values of dogs in Group 3 were the highest, they did not significantly differ from those of the healthy population (Group 1); it is therefore possible to speculate that future studies should include a higher number of individuals in order to confirm the results achieved here.

A sufficient amount of saliva is necessary to protect the oral tissue. The balance between demineralization and remineralization depends on the salivary calcium and phosphate concentration, as well as on the level of salivary alkaline phosphatase [28]. In a human study by Shetty, calcium levels in saliva appeared to decrease correspondingly to an increase in gingival and periodontal inflammation [19]. This is not in accordance with the observations of this study, in which salivary calcium concentration was unaffected by level of periodontal disease. On the contrary, the salivary phosphorus level increased with the increase in the severity of periodontal disease, and ROC curve analysis permitted the affirmation that values above 4.04 mg/dl could be considered predictive of periodontal disease in dogs with moderate accuracy. The significant increase in the phosphorus trend is in agreement with the observations of Shetty, even if in the latter case phosphorus level variation between healthy patients and those with periodontal disease was found to be of no statistical significance. It might be worth to consider that in our study the phosphorus significant *p*-value was quite weak (*p* = 0.049) and this suggests that further studies are required to reinforce the relationship between salivary phosphorus and oral disease in dogs. The existence of a direct relationship between the calcium and phosphorus level of plaque and saliva may be possible, even if it is unlikely that calcium and phosphate ions diffuse from the saliva into plaque [19]. These ions are probably incorporated during the formation of plaque together with salivary proteins. Shetty also hypothesized that the precipitation of calcium and phosphorus may be influenced by the change in pH during plaque formation. In this canine study however, pH was not assessed, and for such reason no further speculation can be made [19].

In human patients, a significant reduction in clinical parameters of periodontal disease has been observed after treatment, accompanied by decreases in salivary amylase and mucin [12]. These results support the hypothesis that salivary glands respond to periodontal disease and increase the protective action of saliva by secreting the non-immunological defenses of the oral cavity, and the secretion rate decreases after the disease disappears [12]. Our study showed lower alpha-amylase concentration in the healthy dogs, even if it did not differ significantly from the values registered in dogs with mild or severe periodontal disease.

Lysozyme is known to interfere on bacterial adherence to oral surfaces, to destroy microorganisms by activating bacterial autolysins that act on their cell walls, and to enhance bacterial clearance from the oral cavity. No previous veterinary medicine studies have measured salivary lysozyme concentrations in dogs with oral disease: although an increasing trend was observed in our study, no significant differences were identified between healthy dogs and those with mild or severe oral disease.

Limitations of the study—A larger sample size could have been recruited to confirm the results obtained in the present study and to ensure that they could be generalized to a larger population. As prior research studies on canine salivary biochemistry are limited, this may have influenced the canine population size included here. A further limitation of the study is represented by the fact that dogs were not sedated prior to oral examination. Although the latter circumstance precluded the evaluation of oral cavity and ligament conditions, it increased owner compliance, allowing a higher number of both healthy client-owned dogs and others with naturally occurring periodontal disease to be included in a reasonably short time.

## 5. Conclusions

This was a pilot study with the main objectives of quantifying concentrations of salivary biochemistry in an unanesthetized client-owned dog population affected by plaque, supragingival calculus, and gingivitis and verifying whether these parameters are affected by grade of periodontal disease. Taking some limitations of the study into account, and considering that, unlike what has been observed in humans, the literature lacks information on the salivary biochemistry in dogs, analyzing salivary calcium, phosphorus, LDH, alpha-amylase, and lysozyme in a canine population enabled us increasing the understanding of the functional role of saliva and its relationship to oral health in this species. Additionally, salivary proteomics could be considered a promising approach to the discovery of other biomarkers potentially linked to the disease.

## Figures and Tables

**Figure 1 animals-12-01091-f001:**
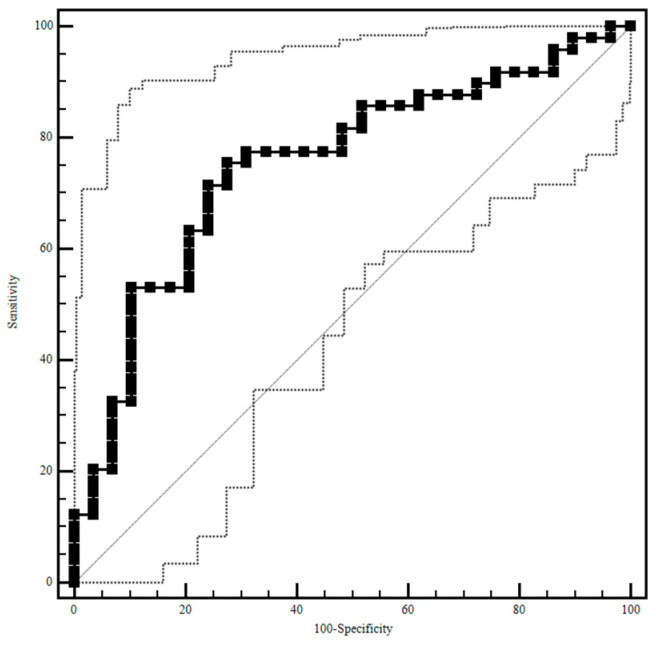
Salivary phosphorus ROC curve analysis.

**Table 1 animals-12-01091-t001:** Scoring criteria of Plaque, Calculus, and Gingivitis index from 0 to 4.

	Plaque Scoring	Calculus Scoring	Gingivitis Scoring
0	no detectable plaque	no detectable calculus	no gingivitis
1	scattered plaque covering less than 24% of the buccal tooth surface	scattered calculus covering less than 24% of the buccal tooth surface	incipient or very mild gingivitis
2	plaque covering between 25% and 49% of the buccal tooth surface	calculus covering between 25% and 49% of the buccal tooth surface	mild gingivitis
3	plaque covering between 50% and 74% of the buccal tooth surface	calculus covering between 50% and 74% of the buccal tooth surface	moderate gingivitis
4	plaque covering more than 75% of the buccal tooth surface	calculus covering more than 75% of the buccal tooth surface	severe gingivitis

**Table 2 animals-12-01091-t002:** Groups 1, 2, 3 and related Plaque, Calculus, and Gingivitis indexes.

Group	Plaque Index	Calculus Index	Gingivitis Index
1	0	0	0
2	1–2	1–2	1–2
3	3–4	3–4	3–4

**Table 3 animals-12-01091-t003:** Descriptive statistics of the population enrolled in the study according to grade of periodontal disease (Group 1, Group 2, Group 3).

	Number of Subjects			
	Group 1	Group 2	Group 3	Total
No. of dogs	29	15	35	79
Age (years)				
≤2	26	3	0	29
2–7	3	10	12	25
>7	0	2	23	25
Age (mean ± sd)	1.5 ± 0.4	4.6 ± 0.6	8.6 ± 0.4	5.3 ± 4.0
Gender				
F/M	16/13	10/5	14/21	40/39
Sterilization				
Yes/No	13/16	7/8	28/7	48/31
Dog size				
Small	12	8	11	31
Medium	11	6	18	35
Large	6	1	6	13
Body Condition Score (mean ± sd)	2.9 ± 0.32	3 ± 0.92	2.91 ± 0.37	2.95 ± 0.50

**Table 4 animals-12-01091-t004:** Salivary biochemistry (mean ± standard error) in healthy dogs and dogs with periodontal disease.

	Group 1	Group 2	Group 3	*p*-Value
Calcium (mg/dL)	7.76 ± 0.75	7.55 ± 0.66	7.98 ± 0.60	0.89
Phosphorus (mg/dL)	3.02 ± 0.76 ^b^	5.34 ± 0.67 ^a^	5.85± 0.61 ^a^	0.049
Lysozyme * (U.I./L)	1.18 (0.62–2.27)	2.01 (1.13–3.57)	2.44 (1.44–4.12)	0.365
LDH (U.I./L)	2559.85 ± 676.95 ^ab^	1636.76 ± 597.36 ^b^	4099.18 ± 545.45 ^a^	0.016
Amylase * (U.I./L)	18.64 (10.05–34.60)	35.44 (21.59–58.22)	27.26 (17.00–43.70)	0.261

Different letters mean differences at *p* values < 0.05. * Lysozyme and amylase values are reported after anti-log transformation and 95% CI within parentheses.

**Table 5 animals-12-01091-t005:** Salivary biochemistry (mean ± standard error) in dogs of different classes of age (≤2 years; 2–7 years; >7 years).

	≤2 Years	2–7 Years	>7 Years	*p*-Value
Calcium (mg/dL)	7.70 ± 0.71	7.19 ± 0.50	8.41 ± 0.74	0.266
Phosphorus (mg/dL)	5.42 ± 0.72	4.73 ± 0.51	4.05 ± 0.75	0.537
Lysozime * (U.I./L)	2.74 (1.67–4.49)	1.14 (0.67–1.90)	1.50 (0.83–2.71)	0.105
LDH (U.I./L)	2643.84 ± 647.62	2594.10 ± 454.77	3057.86 ± 668.89	0.802
Amylase * (U.I./L)	32.14 (18.05–57.22)	20.45 (13.62–30.71)	27.40 (15.54–48.29)	0.360

*p* values < 0.05 are considered significant. * Lysozyme and amylase values are reported after anti-log transformation and 95% CI within parentheses.

**Table 6 animals-12-01091-t006:** Salivary biochemistry (mean ± standard error) in female and male dogs.

	Female	Male	*p*-Value
Calcium (mg/dL)	7.02 ± 0.39	8.52 ± 0.40	0.0078
Phosphorus (mg/dL)	4.33 ± 0.40	5.14 ± 0.41	0.150
Lysozyme * (U.I./L)	1.98 (1.41–2.79)	1.63 (1.14–2.32)	0.415
LDH (U.I./L)	2796.64 ± 355.60	2733. 89 ± 371.96	0.901
Amylase * (U.I./L)	22.57 (16.68–30.55)	30.44 (22.39–41.39)	0.159

*p* values < 0.05 are considered significant. * Lysozyme and amylase values are reported after anti-log transformation and 95% CI within parentheses.

## Data Availability

The data sets supporting the results of this article are included in the article.

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
