# Peer review of "Evaluation of Salivary Biochemistry in Dogs with and without Plaque, Calculus, and Gingivitis: Preliminary Results"

_animals, 2022, doi:10.3390/ani12091091_

Round 1
Reviewer 1 Report
Line 57: “…been identified date in normal and diseased mouths…”. Please delete date
Line 79: “…to plaque formation ([13,14]” »»» “…to plaque formation [13,14]”.
Line 111: “…of various ages, breed…”
Line 128: If in line 128 the authors say “as described in Table 2…” it is not justified to refer to Table 2 in line 130.
Line 140: If the authors do it for the blood, then they must also indicate the centrifugation conditions for the saliva.
Line 220: I think this sentence should appear at the end of the section and added the topic I suggest: “No correlation was found between the analytes quantified in saliva and the same ones measured in serum (data not shown)”. On the other hand, it should be written explicitly that the hematological analytes and serum biochemical measurements were used to identify unhealthy dogs and, more importantly, that the data are not shown, or that they would eventually be used as supplementary material. Excuse me to insist on this point but if the authors don't write it, there's always the doubt why these parameters are determined and then they never appear. Just write "data not shown" or, preferably, "Supplementary Material" and present these Results.
Line 238: “…in saliva ere…” »»» “…in saliva were…”
Line 341: In addition to a broader population to be involved in the future, I suggest other approaches, namely salivary proteomics for the discovery of other biomarkers possibly associated with the disease.
Author Response
REVIEWER 1
Line 57: “…been identified date in normal and diseased mouths…”. Please delete date
Done
Line 79: “…to plaque formation ([13,14]” »»» “…to plaque formation [13,14]”.
Done
Line 111: “…of various ages, breed…”
Done
Line 128: If in line 128 the authors say “as described in Table 2…” it is not justified to refer to Table 2 in line 130.
Done
Line 140: If the authors do it for the blood, then they must also indicate the centrifugation conditions for the saliva.
The reviewer is right and we added the requested detail.
Line 220: I think this sentence should appear at the end of the section and added the topic I suggest: “No correlation was found between the analytes quantified in saliva and the same ones measured in serum (data not shown)”. On the other hand, it should be written explicitly that the hematological analytes and serum biochemical measurements were used to identify unhealthy dogs and, more importantly, that the data are not shown, or that they would eventually be used as supplementary material. Excuse me to insist on this point but if the authors don't write it, there's always the doubt why these parameters are determined and then they never appear. Just write "data not shown" or, preferably, "Supplementary Material" and present these Results.
We agree with the reviewer. As suggested, the sentence “No correlation was found between the analytes quantified in saliva and the same ones measured in serum (data not shown)” has been moved at the end of the Results-section.
Furthermore, we modified the first sentence of Materials and methods as follows: “Dog population – for inclusion in this study a general physical examination and the hematological analytes and serum biochemical measurements within physiological ranges were used to identify healthydogs.”. The expression “data not shown” related to theblood parameters was added in the first sentence of the Results-section. We hope that you are agree with these proposals.
Line 238: “…in saliva ere…” »»» “…in saliva were…”
Done
Line 341: In addition to a broader population to be involved in the future, I suggest other approaches, namely salivary proteomics for the discovery of other biomarkers possibly associated with the disease.
Thank you for your kind suggestion. We added the sentence in the text.

Reviewer 2 Report
It seems that the manuscript did not undergo many significant changes and improvements with respect to the previous version
Author Response
It seems that the manuscript did not undergo many significant changes and improvements with respect to the previous version
We are sorry that we were not able to satisfy you expectancies. Although we are aware that this study has some limitations, we are confident that we added knowledge in a field which is few investigated in veterinary medicine. For this reason more research are needed on this topic and we look forward to seeing our preliminary results to be confirmed by other similar research studies.

Reviewer 3 Report
Dear Autors,
Did line 26-27 appear in the manuscript by accident?
The manuscript has undergone a thorough reconstruction.
I reiterate my latest rating of this article. I still think that it is an interesting research topic. The use of non-invasive biological material is important in minimizing stress on animals in the veterinary clinic.
Author Response
Did line 26-27 appear in the manuscript by accident?
Thanks, you are right: it was a mistake. That piece of sentence has been deleted.
The manuscript has undergone a thorough reconstruction. I reiterate my latest rating of this article. I still think that it is an interesting research topic. The use of non-invasive biological material is important in minimizing stress on animals in the veterinary clinic.
Thanks for your comment.

This manuscript is a resubmission of an earlier submission. The following is a list of the peer review reports and author responses from that submission.
Round 1
Reviewer 1 Report
The aim of this study was to quantify the salivary concentration of α-amylase, lysozyme, lactate dehydrogenase, calcium and phosphorus in a conscious dog population affected by plaque, supragingival calculus and gingivitis and to evaluate whether these parameters can be used as predictive markers of periodontal disease in this species.
INTRODUCTION
Comment 1 (line 68): bio markers »» biomarkers
Comment 2 (line 73): LDH »» lactate dehydrogenase (LDH)
MATERIALS AND METHODS
Comment 3 (lines 97): Although the authors refer “…moreover they had to be fed commercial dry pet food”, were the owners asked about the diet/food quality and possible supplementation to prevent dental problems?
Comment 4 (lines 105-106): “…as described by Warrick et al. (2000)…” »» “…as described by Warrick et al. [5]…”
Comment 5 (lines 108-109): “…evaluations in non-sedated animals (Warrick et al., 2000)” »» “…evaluations in non-sedated animals [5]”
Comment 6 (line 119): In Table 2, Group 2, Gingivitis Index is 1-2 or 1-3?
Comment 7 (line 120): Were the collections made at the same time of day?
Comment 8 (line 121): “…reported by Iacopetti et al. (2017)” »» “…reported by Iacopetti et al. [15]”
Comment 9 (line 126): I suggest separating these two paragraphs.
Comment 10 (line 126): How was the blood collected and processed?
Comment 11 (Saliva and blood collection and analysis): Although the authors say that "Both saliva and blood samples were collected, processed and analyzed as previously reported by Iacopetti et al. (2017)", at least refer the main techniques.
Comment 12 (line 148): Please confirm the References, Bland and Altman, 1996, does not appear in the bibliography.
RESULTS
Comment 13: Why are the results of all determinations not shown? In Tables only appear Calcium, Phosphorus, Lysozyme, LDH,Amylase. This aspect should be clarified.
“The hematological analytes included packed cell volume (PCV), platelet count (PLT), and leukocyte count (WBC), as well as the relative and absolute numbers of neutrophils, lymphocytes, monocytes, eosinophils, basophils, and large unstained cells (LUC) counts, and values for MPV, large platelet count, and platelet clumps (aggregates). The serum biochemical analytes included:…” On the other hand in line 135 the authors refer that “Further, the analytes quantified in saliva were also measured in the serum”. Have any correlations been determined?
Comment 14 (Lines 178): Table 4 » The table title should just be “Salivary biochemistry (mean±standard error) in healthy dogs and dogs with periodontal disease. The other text should appear in a footnote “P values <0.05 are considered significant. * For lysozyme and amylase, values are reported after anti-log transformation and 95% C.I. within parentheses”.
Comment 15 (lines 194 and 197): Table 5 and Table 6, respectively » The same for the titles of these tables as for Table 4.
Reviewer 2 Report
Evaluation of salivary biochemistry in dogs with and without 2 plaque, calculus and gingivitis: preliminary results.
Interesting research topic. The use of non-invasive biological material is important in minimizing stress in animals in a veterinary clinic. Therefore, the research carried out by the authors is innovative and has great application importance.
Abstract:
In my opinion, the abstract subheadings (background, methods, results, conclusion) are redundant. Such a division results from the content of the abstract itself.
Material and methods:
In my opinion, it would be better to start describing the research group about the criteria that the dogs had to meet. Only later should it be written that dogs of different breeds, ages, etc. took part in the study. Simply change the order. The first sentence in this paragraph simply suggests to the reader that the group was sort of randomly selected. But that's just my cosmetic remark.
Line 124-125: Two cotton swabs and two test tubes were used to obtain more saliva for analysis? I guess so. It would be good to explain this in the text.
Results:
The most important results are detailed in the tables.
I wonder about the sense of describing what analyzes were collected from blood (in Material and methods). In the chapter describing the results, the results of the hematological analyzes are not presented. So maybe it would be better to focus only on saliva as the biological material used in this research?
Discussion:
A thorough analysis of the obtained results with the results of other authors
Reviewer 3 Report
This manuscript has major limitations:
*The objective seems not clear: In first sentence of the discussion is indicated “prognostic tool”. These analytes are evaluated for prognosis?
*The age of the groups evaluated should be similar in order to do a proper comparison. This is an important point.
*The assays should be more deeply described and data of their validation provided.
*The number of animals is not enough to establish differences between males and females.
*There is no clear explanation to pass from groups to 0 to 4 (5 groups) to only 3 groups.
*The data of sensitivity and specificity of around 70% for P is not enough to give such importance to P for the detection of the disease.
*The discussion is highly speculative, the only analyte that could show a potential is P, why the rest of analytes are so deeply discussed?
*The results of LDH with lower values in the second group of animals is strange, possible the low number of animals used and the different ages would be the responsible of this.
Minor comments:
-English should be reviewed
-A general evaluation of the buccal health is not a so painful procedure and can be done in most dogs except the aggressive ones